# Identification of Gene Expression in Different Stages of Breast Cancer with Machine Learning

**DOI:** 10.3390/cancers16101864

**Published:** 2024-05-14

**Authors:** Ali Abidalkareem, Ali K. Ibrahim, Moaed Abd, Oneeb Rehman, Hanqi Zhuang

**Affiliations:** 1EECS Department, Florida Atlantic University, Boca Raton, FL 33431, USA; aabidalkaree2015@fau.edu (A.A.); orehman@fau.edu (O.R.); zhuang@fau.edu (H.Z.); 2Harbor Branch Oceanographic Institute, Florida Atlantic University, Fort Pierce, FL 34946, USA; 3Ocean and Mechanical Engineering Department, Florida Atlantic University, Boca Raton, FL 33431, USA; mabd2015@fau.edu

**Keywords:** machine learning, breast cancer, microRNA, cancer stage classification, bio-marker identification

## Abstract

**Simple Summary:**

Metastatic breast cancer is an aggressive disease that early diagnostic attempts is of an utmost importance. A machine learning model that utilizes NCA and MRMR in this work is attempting to isolate pertinent dysregulated miRNA’s for the different four cancer stages. This work compares the current clinical diagnostic approaches with the proposed ML model results.

**Abstract:**

Determining the tumor origin in humans is vital in clinical applications of molecular diagnostics. Metastatic cancer is usually a very aggressive disease with limited diagnostic procedures, despite the fact that many protocols have been evaluated for their effectiveness in prognostication. Research has shown that dysregulation in miRNAs (a class of non-coding, regulatory RNAs) is remarkably involved in oncogenic conditions. This research paper aims to develop a machine learning model that processes an array of miRNAs in 1097 metastatic tissue samples from patients who suffered from various stages of breast cancer. The suggested machine learning model is fed with miRNA quantitative read count data taken from The Cancer Genome Atlas Data Repository. Two main feature-selection techniques have been used, mainly Neighborhood Component Analysis and Minimum Redundancy Maximum Relevance, to identify the most discriminant and relevant miRNAs for their up-regulated and down-regulated states. These miRNAs are then validated as biological identifiers for each of the four cancer stages in breast tumors. Both machine learning algorithms yield performance scores that are significantly higher than the traditional fold-change approach, particularly in earlier stages of cancer, with Neighborhood Component Analysis and Minimum Redundancy Maximum Relevance achieving accuracy scores of up to 0.983 and 0.931, respectively, compared to 0.920 for the FC method. This study underscores the potential of advanced feature-selection methods in enhancing the accuracy of cancer stage identification, paving the way for improved diagnostic and therapeutic strategies in oncology.

## 1. Introduction

According to the World Health Organization [1], 2.1 million women are affected by breast cancer each year. Among women, breast cancer is the most common cause of cancer-related mortality (15% in 2018), with higher disease incidence observed in developed countries such as the United States and the United Kingdom. The causation of breast cancer can be multifactorial and involve factors such as reproductive age [2], age at onset of menopause [3], contraceptive use [4], hormonal therapy [5], and exogenous hormones used for in vitro fertilization [2]. Additionally, breast tumors can arise from hereditary gene mutations, commonly associated with BRCA1 and BRCA2 genes [2]. These tumors exhibit heterogeneity and can be classified using various prognostic biomarkers, including proto-oncogene expression [6], growth factor receptors, inflammatory cytokine levels [7], and microRNA (miRNA) expression [8]. Dysregulation and differential expression of miRNAs in breast cancer provide a basis for studying its etiology as well as identifying novel prognostic markers.

miRNA is a class of non-coding RNA that is evolutionary conserved and is expressed in many animals and plants. Made of approximately 20–22 nucleotides, miRNAs are known to decrease gene expression [9], mediate protein expression [10], and upregulate translation of targeted mRNA [10,11]. Most miRNAs are transcribed in the nucleus as long primary transcripts (pri-miRNAs) via RNA polymerase II. Drosha-DGCR8 complex process pri-miRNAs to a stable 70-nucleotide stem-loop RNA (pre-miRNA) which is exported by Exportin 5 to the cytoplasm. Dicer proteins cleave the pre-miRNAs to 20–22 nucleotide RNA duplex strands. Strands that have a base pairing at the 5’ end act as the mature miRNA, while passenger strands are degraded [12]. The mature miRNA guide argonautes to form an RNA-induced silencing complex (miRISC) which binds as a partial complement strand to targeted mRNA leading to suppression and regulation of many protein-coding genes [13]. Thus, miRNAs play a pivotal role in cellular processes such as proliferation, differentiation, and apoptosis [13]. However, aberrant miRNA expression is a hallmark of diseases such as cancer. It is still unclear on how miRNAs become dysregulated in cancer. However, studies have shown that many human miRNA loci are located in cancer-associated genomic regions [14]. Therefore, a number of classifiers have been developed as prognostic tools to identify and analyze expression signatures of miRNA in breast tumors.

Machine learning methods have been used to identify MicroRNA as biomarkers in cancer stage progression, yielding promising results in enhancing the accuracy and efficiency of cancer diagnostics. This trend extends across various medical domains, including evaluating hospital efficiency for stroke care, assessing technical efficiency in public hospitals during and after the COVID-19 pandemic, developing molecular classification models for triple-negative breast cancer subtypes, and devising strategies for automated diagnosis and personalized treatment [15,16,17,18]. These diverse applications underscore the significant role of machine learning in improving diagnostic precision, operational effectiveness, and personalized medical care.

In this research, we focus on breast cancer stage classification using miRNAs as biomarkers, which was not adequately studied in the literature. We have chosen the Neighborhood Component Analysis (NCA) and Minimum Redundancy Maximum Relevance (MRMR) techniques to identify the most impactful miRNAs in each breast cancer stage. This choice is driven by several critical factors. Features selected through the MRMR technique are not only highly relevant for targeting the variable of interest but also exhibit minimal redundancy. This reduction in redundancy is crucial in the complex, overlapping the world of genetic data. Furthermore, the NCA technique preserves the local structure of the data of interest. This characteristic is essential for understanding nuanced relationships and proximities between genes. Additionally, experimental studies have demonstrated that both techniques are computationally efficient, making this approach time-efficient for processing multi-dimensional data. The focused application of the NCA technique and the global feature relevance provided by MRMR offer a comprehensive technical approach for the feature selection process.

This work is a continuation of our previous work [19,20,21]. In this research, we compare our results with the results obtained using the fold-change approach. The remainder of the paper is as follows. The architecture of the miRNA biomarker identification system is proposed in Section 3 along with the machine learning algorithms used for this study. Experimental design and results are given in Section 4. The paper ends with concluding remarks in Section 6.

## 2. Literature Review

In 2002, the first case of aberrant miRNAs was identified as a cluster of miR-15 and miR-16 at chromosome 13q14, a commonly deleted region in chronic lymphocytic leukemia [22]. Since that case, many advances have been made to identify and characterize miRNAs in the role of all cancer hallmarks [23] as well as their implication in clinical management at every stage of cancer. Initial identification of miRNAs has been successfully facilitated with broad-scale expression profiling. The methods used for miRNA profiling include microarrays [24], high-throughput deep sequencing [25,26], and bead-based flow cytometric miRNA analysis [27]. Sorlie’s et al. groundbreaking work utilized microarrays to generate gene expression signatures that designated subtypes of squamous breast cancer based on estrogen receptor (ER), progesterone (PR), and HER2/neu receptor status [28,29]. Linear amplification via qRT-PCR and microarrays for pictogram quantities of miRNA were used by Mattie et al. to characterize novel sets of miRNAs defined by HER2/neu or ER/PR receptor status [30]. Microarray analyses used by Iorio et al. identified 29 miRNAs that were abnormally expressed in breast tumors in addition to identifying 15 miRNAs that could be used to differentiate tumor tissue from normal tissue [31].

The human epidermal growth factor receptor (HER) family of receptors plays a critical role also in the prognostication process of several human cancer types [32]. Tumorgenesis and cell proliferation of approximately 15–30% of breast cancer have been linked to human epidermal growth factor receptor 2 (HER2) [33]. However, the significance of HER2 overexpression in patients with ductal carcinoma in situ (DCIS) remains poorly defined [34]. DCIS is a heterogeneous disease, and the clinical significance recurrence rates have been reported after breast-conserving surgery for DCIS patients. With the widespread screening technologies like mammography, early-detected cases of DCIS are on the rise. Early predictive models for breast cancer progression can assist clinicians in selecting timely treatment schemes for those patients.

In comparison to running miRNA expression profiles on invasively collected breast biopsies, minimally invasive direct serum assays have proven to be a sensitive diagnostic approach to identifying circulating miRNAs. For example, Asaga et al. applied RT-qPCR directly to serum collected from 102 patients with varying stages of breast cancer [35]. This method was successfully used to detect circulating miR-21, one of the most dramatically up-regulated miRNAs in breast cancer that is associated with tumor progression, metastasis, and poor prognosis [35,36]. A study carried out by Haman et al. enhanced the isolation of circulating miRNA by 5-fold from 23 breast cancer patients by using speed-vacuum concentration [37]. This technique was combined with global profiling which is advantageous compared to qRT-PCR since novel miRNA expression can be elucidated from different breast cancer stage types [37]. Additionally, individual patient data analysis can be conducted with global profiling to assess miRNA expression rather than using pooled samples from multiple patients for qRT-PCR [37,38]. Thus, various diagnostic and prognostic strategies have been developed for identifying miRNA in the implication of breast cancer characterization and progression. The key role of miRNA in identifying and diagnosing breast cancer motivated studies in identifying potential miRNA targets and investigate therapeutic procedures.

Lu et al. showed that microRNA expression profiles are effective in classifying human cancers, offering a viable approach for diagnosing cancer and identifying its subtypes [27]. The study conducted by [39], using KNN and decision trees, analyzed 400 paraffin-embedded and fresh-frozen samples from 22 different tumor tissues and metastases, achieving high confidence with an accuracy of 90% on two-thirds of those samples. It also confirmed the role of miRNAs as biomarker agents in oncogenic diseases. The research carried out by [40] explored datasets of liver, lung, and brain cancers to identify genuine miRNA biomarkers for diagnosing these diseases. Techniques such as the Support Vector Machine (SVM) and an ensemble of specific filters were used to detect carcinogenic agents. Building on this, another article reviews various machine learning methods for classifying cancer types based on gene expression data, assessing their accuracy, effectiveness, and clinical applicability [41]. The authors of [42,43] analyzed 70 matched pairs of intact renal cell carcinoma and normal kidney tissues and identified 166 miRNAs which were substantially dysregulated in clear cell and renal cell carcinoma. As data increased over the years and new machine learning (ML) techniques advanced, more sophisticated classification approaches have been used.

The Cancer Genome Atlas Program (TCGA), which was a collaboration project between the National Cancer Institute and the National Human Genome Research Institute, generated more than 2.5 petabytes of genomic, epicgenomic, tanscriptomic, and proteomic data. The dataset is publicly available and contains primary cancer and matched normal samples for more than 30 cancer types. In [44], a random forest classifier was used to classify metastasis status using gene expression data from TCGA. Moreover, the proposed approach assigned a metastasis score to each gene to identify important genes. A Deep Learning approach was used to analyze RNA sequencing (RNA-seq) gene expression data for the purpose of classifying various types of cancer [45]. This analytical capability is critical as it potentially leads to more precise cancer diagnostics and the development of targeted therapies, thereby advancing the field of oncology by improving both the accuracy and efficiency of cancer classification.

High dimensionality in biomedical data led to the introduction of numerous approaches to salve the problem, though developing a robust predictive model that takes into the account computational cost, accuracy, and explanability has been a challenge in the field of biomedical research. A Rat Swarm Optimizer is proposed as a feature selection technique that finds the most representative data from a given dataset. This feature selection techniques is based on three successive modifications: the s-shape transfer function that is used to develop the RSO algorithms, the local search paradigm of particle swarm optimization, and three crossover mechanisms that are used and controlled by a switch probability to improve the diversity [46]. Awadallah and their coauthors developed the Binary Horse Herd Optimization Algorithm (BHOA) to address the high dimensionality problem. The BHOA algorithm incorporated the following two adjustments: three transfer functions are used to transform the continuous domain into a binary domain, and three crossover optimizers were used to enhance the efficiency of the BHOA [47]. Yaqoob’s group conducted a systematic review of the most nature-inspired algorithms, such as crow search algorithms, ant lion optimizer, and moth flame optimization, to tackle the problem of high dimensionality of biomedical data for the purposes of predication and classification [48].

Recently, different studies have focused on miRNA expression profiling in differentiating the stages of breast cancer, indicating its potential as a biomarker for both the presence and progression of the disease. One study analyzed miRNA expressions using arrays in various stages and grades of invasive ductal carcinoma (IDC), revealing significant differences in expression across stages, which could assist in understanding disease progression [49]. Another study focused on mouse models, investigating miRNA–mRNA interactions throughout different stages of cancer development, showing how miRNA expression correlates with cancer progression stages [50]. Moreover, research into the clinical association of miR-10b with breast cancer stages used real-time PCR to demonstrate that varying levels of miR-10b are statistically significant and correlate with advanced disease stages, underlining the role of miRNAs in stage-specific regulation and their potential in guiding targeted therapies and prognostic assessments [51].

In this research, a significant gap is identified in the application of advanced machine learning techniques to utilize miRNAs as biomarkers for breast cancer stage classification. Despite the recognized role of miRNAs in the pathogenesis and progression of cancer, the specific use of NCA and MRMR for identifying microRNA as biomarkers between stages of breast cancer remains underexplored. This gap is notable because these techniques offer the potential to significantly improve diagnostic accuracy by effectively handling the complex, high-dimensional genetic data characteristics of cancerous tissues. Addressing this gap could lead to select targeted therapeutic strategies and improved patient outcomes.

## 3. Methodology

The proposed model for the stage identification of the breast cancers using miRNA as biomarkers is shown in the following Figure 1. The proposed approach begins with the extraction Breast Cancer data from the TCGA database. We downloaded the cancer data from the TCGA server, and we categorized the data using MATLAB programming. Data categorization starts by organizing each case ID with its respective miRNA quantification data, using unique identifiers. The study primarily concentrates on the levels of miRNA, quantified as reads per million. Subsequently, these miRNA quantifications are associated with clinical data by matching case IDs and File IDs. The clinical data are stored in the format of JSON file, containing detailed information on cancer stage and pertinent patient demographics such as age and gender. After matching each miRNA file with stage information, features selection was applied to examine the gene regulation aberration of diseased tissue against the healthy ones for each indicated stage. The labels of the samples with quantitative miRNAs were provided by TCGA. This process is shown in the ‘Labeling’ stage of the diagram. Two feature selection methods, namely NCA and MRMR, were employed in the study to identify and isolate discriminative biomarkers that are most influential to each associated stage. Finally, SVM is used as the computation method of choice to classify the features into four different stages. No effort is made in this study to optimize the classifier, since the goal of the the research is to pinpoint miRNAs as biomarkers for cancer stage identification. For readability, the NCA and MRMR feature selection algorithms are discussed next.

### 3.1. Gene Expression Data Analysis

Fold-change is a commonly used statistical technique in gene expression analysis which measures the difference in expression levels between two conditions, typically a treatment group and a control group. It is calculated as the ratio of the expression level of a gene in the treatment group to that in the control group. For example, if the expression level of a gene in the treatment group is twice that of the control group, the fold-change is 2 [52,53].

Fold-change is often reported in a logarithmic scale of base two, which means that a two-fold change is equivalent to a log2 value of 1, a four-fold change is equivalent to a log2 value of 2, and so on.

The prevalence of the fold-change technique in gene expression analysis such as transcriptomics, proteomics, and metabolomics, for measuring the changes in different conditions, as opposed to some standard value is the reason of this technique being chosen in this paper and compared to our experimental model that uses modern computational techniques. Deferentially expressed genes in this paper are defined to be miRNA data that are statistical outliers from some standard state. The differential expressions for example in Figure 2 compares the average expression of a gene in group A with the average expression of the same gene in group B. The fold-change method will yield a positive fold-change to indicate there is an increase in the expression. Comparatively, it will report a negative fold-change if their is a decrease in the expression for that gene. The value is typically reported in a logarithmic scale of base two. The p-value is also used as a statistical indicator for the likelihood of a gene to be deferentially expressed in this method. By incorporating the *p*-value, we aim to determine whether the observed changes in gene expression are likely to be due to random variation or are statistically significant. Specifically, the *p*-value helps in testing the null hypothesis that there is no difference in gene expression between the treatment and control groups. A low *p*-value (typically less than 0.05) indicates that the observed fold-change is unlikely to have occurred by chance, thereby supporting the alternative hypothesis that there is a significant difference in expression. This statistical approach ensures that the identified differentially expressed genes are not only different in terms of the fold-change but are also significant statistically, minimizing the risk of false positives.

### 3.2. Neighborhood Component Analysis

NCA is a non-parametric feature selection method that can be used to extract multivariant features to maximize prediction performance for supervised machine learning classification and regression algorithms. The goal of NCA is to minimize the objective function that is responsible to measure the average leave-one-out (LOO) performance for classification or to measure the regression loss for the data that are used for training [20,54]. This can be used for classification tasks and can be described as follows:

Let $S=\left(〖miRNAs{〗}_{i},〖stages{〗}_{i}\right)$, where i=1,2,3,……,n, and $miRNAs_{i}\in S$ are the feature vector (1881 miRNAs), and $stages_{i}\in 1,2,3,4$ are cancer class labels. The main idea of the NCA is to find a model f:miRNAs→stages that accepts miRNAs as features and outputs predictions for different stages. To build this model, consider the following ransomized classifier that
Randomly picks a point, Ref(miRNAs), from *S* as the reference point for a given miRNA.Labels miRNAs using the label of the reference point Ref(miRNAs).

All other points will have some probability of being selected as a reference point. The probability $P(Ref\left(miRNAs\right)=〖miRNA{〗}_{j}S$) that point $〖miRNA{〗}_{j}$ is picked from *S* as the reference point for miRNA is higher if $〖miRNAs{〗}_{j}$ is closer to miRNA as measured by the following distance function:(1)$$d_{w}\left(miRNA_{i},miRNA_{j}\right)=\sum_{r=1}^{p}w_{r}^{2}miRNA_{ir}-miRNA_{jr}$$
where $w_{r}$ are weights of miRNAs. To calculate the weight for each miRNA, let us assume that $P\left(Ref\left(miRNA\right)=〖miRNA{〗}_{j}S\right)\propto g(d_{w}\left(miRNA,〖miRNA{〗}_{j}\right)$, where *g* is a kernel function or a similarity function that maps small distance metrics $d_{w}\left(miRNA,〖miRNA{〗}_{j}\right)$ to another value, usually a larger value. Let us also assume that
(2)$$g\left(z\right)=e^{\frac{-z}{\sigma }}$$

Since the reference point miRNA is chosen from *S* and the sum of all probabilities shall add up to 1, we can write the following equation:(3)$$P\left(Ref\left(miRNA\right)=miRNA_{j}/S\right)=\frac{d_{w}\left(miRNA,miRNA_{j}\right)}{\sum_{j=1}^{n}\left(d_{w}\left(miRNA,miRNA_{j}\right)\right)}$$

The training set *S* excluding the point $\left(miRNA_{i},Stage_{i}\right)$ includes data in $S^{-i}$ of this randomized classifier in which the leave-one-out application is being used. To calculate the probability that a $miRNA_{j}$ is chosen as a reference point for $miRNA_{i}$, the following equation can be used:(4)$$P_{ij}=P\left(Ref\left(miRNA_{i}\right)=miRNA_{j}/S^{-i}\right)=\frac{d_{w}\left(miRNA_{i},miRNA_{j}\right)}{\sum_{j=1}^{n}\left(d_{w}\left(miRNA_{i},miRNA_{j}\right)\right)}$$

The probability of the average leave-one-out of correct classification is the probability $P_{i}$, which can be computed using the following equation:(5)$$P_{i}=\sum_{j=1,j\neq i}^{n}P\left(Ref\left(miRNA_{i}\right)=miRNA_{j}/S^{-i}\right)I\left(Stage_{i}=Stage_{j}\right)=\sum_{j=1,j\neq i}^{n}p_{ij}stage_{ij}$$
in which
(6)$$stage_{ij}=I\left(stage_{i}=stage_{j}\right)=\left\{\begin{array}{cc} 1, & \mathrm{if}\;stage_{i}=stage_{j}\\ 0, & \mathrm{otherwise} \end{array}\right.$$

Therefore, the correct classification probability of the average leave-one-out using the randomized classifier can be written as
(7)$$F\left(w\right)=\frac{1}{n}\sum_{i=1}^{n}p_{i}$$

The main objective of NCA is to maximize the function F(w) with respect to *w*. A regularization term is introduced to make the optimization algorithm more robust.
(8)$$F\left(w\right)=\frac{1}{n}\sum_{i=1}^{n}p_{i}-\lambda \sum_{r=1}^{n}w_{r}^{2}$$
(9)$$F\left(w\right)=\frac{1}{n}\sum_{i=1}^{n}\left[\sum_{j=1}^{n}p_{ij}stage_{ij}-\lambda \sum_{r=1}^{n}w_{r}^{2}\right]$$
where λ is the regularization parameter. After selecting the kernel parameter σ in $p_{ij}$ as 1, the following minimization equation can be used to find the weight vector *w* for a given value of λ.
(10)$$\hat{w}=argmin_{w}f\left(w\right)=argmin_{w}\frac{1}{n}\sum_{i=1}^{n}f_{i}\left(w\right)$$
subject to following constraints, $\sum_{i=1}^{n}\sum_{j=1,j\neq i}^{n}p_{ij}=1$

If a constant is added to the objective function, the argument of the optimization problem will not change. Hence, a constant can be added to the objective function as shown below.
(11)$$\begin{array}{c} \hat{w}=argmin_{w}\left(1+f\left(w\right)\right)\\ =argmin_{w}\frac{1}{n}\sum_{i=1}^{n}\sum_{j=1,j\neq i}^{n}p_{ij}-\frac{1}{n}\sum_{i=1}^{n}\sum_{j=1,j\neq i}^{n}p_{ij}stage_{ij}+\lambda \sum_{r=1}^{p}w_{r}^{2}\\ =\sum_{i=1}^{n}\sum_{j=1,j\neq i}^{n}p_{ij}\left(1-stage_{ij}\right)+\lambda \sum_{r=1}^{p}w_{r}^{2}\\ =\frac{1}{n}\sum_{i=1}^{n}\sum_{j=1,j\neq i}^{n}p_{ij}I\left(stage_{i},stage_{j}\right)+\lambda \sum_{r=1}^{p}w_{r}^{2} \end{array}$$
where $I(stage_{i},stage_{j})$ is the loss function that is defined by Equation (Equation 6).

### 3.3. Minimum Redundancy Maximum Relevance

The MRMR algorithm computes and identifies an optimal set of features that are mutually and maximally dissimilar and can represent the response variable effectively [55]. This research attempts to identify the most influential miRNAs that associate the five different stages of breast cancer and nominate the culprits as biomarkers for each said stage. This algorithm is selected to discriminate among the miRNAs due to the fact that it minimizes the redundancy and maximizes the relevance of a feature set to the response variable. MRMR is also capable of outputting values that gauge the redundancy itself in the feature set. It uses mutual information of pairwise features and mutual information of a feature and the response. It is, therefore, a desirable tool for the problem that this study attempts to solve.

An optimal set of features *S* shall be chosen by the algorithm to maximize $V_{s}$, the relevance of *S* in correspondence with the response variable stage. The algorithm also solves another optimization problem in which it minimizes $W_{s}$, the redundancy in *S*. $V_{s}$ and $W_{s}$ are expressed in the following equations, respectively:(12)$$V_{s}=\frac{1}{S}\sum_{miRNA\in S}^{I}\left(miRNA,stage\right)$$
(13)$$W_{s}=\frac{1}{S^{2}}\sum_{miRNA,z\in S}^{I}\left(miRNA,z\right)$$

Unlike parsing through all the combinations of $2^{\Omega }$ to find the optimal set features *S*, MRMR ranks features via the forward-addition procedure, which requires O(Ω.S) computations. The Mutual Information Quotient (MIQ) can be calculated using
(14)$$MIQ=\frac{V_{miRNA}}{W_{miRNA}}$$

## 4. Experimental Design and Results

The Cancer Genome Atlas Program offers a vast source of information for numerous human cancer types and their different stages. In this research, 1097 distinct metastatic tissues with their associated stage number were analyzed and used to inject into the proposed breast cancer stage identification system, consisting of the Labeling, Feature Selection and Classification blocks in Figure 1. In the experimental study, 1207 samples were extracted from the 1097 breast cancer patients, 113 of which are considered normal tissue samples taken from unaffected areas of the patient body. In total, 111 diseased-tissue samples were obtained from patients who have stage-one breast cancer, 350 samples were associated with stage-two cases. A total of 131 samples were correlated with stage-3 cancer, and 11 samples belonged to stage 4. Samples that were labeled as Stage X were not included (which accounts only for five samples) in our model. Blood-specific samples were not used in the experimental study because blood samples from patients exhibited the same miRNA composition for both normal tissues and cancer tissues, rendering them ineffective for pinpointing the most impactful miRNAs for breast cancer stage classification. As a result, 716 tissue samples were used in our experiments. The proposed model ran each sub-data miRNAs against normal breast tissues to recognize and isolate the suspected dysregulated genes in the formation of the disease. Due to the recent advantage of machine and deep learning, it has become possible to use machine learning algorithms to study the effect of each gene on the specific kind of cancer. Multiple machine learning approaches were applied for this purpose, and finally, the SVM algorithm was selected due to its simplicity and satisfactory performance. To establish a baseline for comparison with the proposed method, we selected the fold-change (FC) method, a widely used approach among researchers and practitioners. We compared Random Forest and Chi-Square methods with the FC method (the baseline) and found out that they performed worse than the baseline method, and therefore, we did not report these results in the paper. In each stage, we compared miRNA expression between tumors of diseased tissues and normal ones using the FC method. Table 1 shows the captured genes by the proposed model that have been revealed to be the most influential biomarkers in detecting tumor Stage 1. Ten up-regulated miRNAs are listed in breast cancer Stage 1 and ten down-regulated are also chosen to be the differentiators of Stage 1; refer to Table 1. The table also shows the results of the statistical analysis carried out with the FC method. It is worth noting that the adjusted *p* value was needed in the FC method to control the False Positive Rate in such a scenario where multiple-hypothesis testing was generated. The Benjamini–Hochberg method was used to calculate those adjusted *p* values. The Benjamini–Hochberg method is a statistical approach designed to control the FDR in multiple-hypothesis testing scenarios. This method involves ranking the *p*-values from multiple tests in ascending order, setting individual thresholds for each *p*-value based on its rank and the desired FDR level, and identifying the largest *p*-value that falls below its corresponding threshold as significant [56].

Table 2 presents the results for the subsequent stages and the associated biomarkers (including the up-regulated and down-regulated genes) identified with the FC method. The most discriminant dysregulated genes that identify Stage 2, 3, and 4 were aggregated in this table.

The accuracy of the up-regulated and down-regulated genes in identifying their respective corresponding stages are shown in Table 3. The measure of accuracy is subject to the following equation.
(15)$$Accuracy=\frac{TP+TN}{TP+TN+FP+FN}$$

It can be hypothesized from the tables above that the FC method is suggesting that as the cancer stage increases, the gene expressions and their dysregulation varies proportionally for the down-regulated miRNAs and inversely for the up-regulated miRNAs. Analogously, Table 3 reports other sets of differentiated genes for each of those four stages, with their related accuracy shown later.

We also used the statistical method chi-square to identify important miRNAs in each stage. It is a statistical test to determine the dependency of a feature on the class label. We can discard features that do not show dependency and extract the relevant features that are useful for classification. Table 4 presents the important genes for each stage. Furthermore, Figure 3 depicts the relationship between the stage and the number of expressed genes. It is interesting to note that, as cancer progresses to later stages, the number of up-regulated genes decreases, but this trend does not hold for down-regulated genes.

Next, we conducted an evaluation of our machine learning models’ performance in identifying breast cancer stages using a five-fold validation approach. We randomly allocated 80% of the dataset for training, reserving the remaining 20% for testing. This process was iterated five times, ensuring that each sample in the dataset was validated once. The outcomes obtained from these five iterations were subsequently averaged. The SVM Algorithm was used as a classifier (Binary classification) to compute the most influential miRNAs. Table 5 shows the results of this experimentation where the NCA algorithm was used for feature extraction, and the SVM algorithm was used for classification. The results obtained by using MRMR with SVM are shown in Table 6. These computational approaches were compared in terms of their accuracy in detecting breast cancer and its stage, and the results are shown in Table 7. It can be seen that the proposed methods outperformed the other two methods significantly, especially for earlier stages of cancer.

To ensure relevance, we concentrated on the common miRNAs among the top 50 biomarkers identified by each of NCA and MRMR methods for each stage of breast cancer. This strategy allowed us to narrow down our analysis to the most statistically significant and functionally relevant miRNA biomarkers. Table 8 shows the common biomarkers for each stage.

Furthermore, we used these common biomarkers to calculate the accuracy of each stage. Table 9 shows the accuracy of each stage when we used only these common miRNA.

It is evident that the common biomarkers identified by both procedures perform well for stages 2 and 4 but exhibit lower effectiveness for stages 1 and 3. Future research is needed to investigate the reasons behind this behavior of these biomarkers for cancer stage identification.

## 5. Discussion

We experimented our proposed approach with the TCGA breast cancer dataset, which categorized patient samples into four stages of cancer to identify microRNA regulation in each stage. The fold-change (FC) method was used as a baseline to identify microRNA expressions in cancerous versus normal tissues, with statistical adjustments like the Benjamini–Hochberg method ensuring reliability in the face of multiple hypotheses testing. The SVM method was used to improve the ablity to identify patterns in gene expression related to cancer severity and progression. Statistical methods like the chi-square test, combined with the machine learning approach, provide a robust framework for pinpointing significant miRNA markers. The proposed solution assesses the potential of microRNAs as biomarkers across various cancer stages, enhancing the model’s predictive power. Different features selection methods, including neighborhood component analysis (NCA) and maximum relevance minimum redundancy (MRMR), were evaluated using a five-fold cross-validation approach. The models’ performances were compared, highlighting the superior accuracy of NCA in most instances.

The results suggest that advanced feature selection approaches can significantly contribute to the precision of genetic analyses in cancer research. They offer promising pathways for the integration of the Internet of Things in surgical practices, potentially revolutionizing fields like telesurgery and telementoring through enhanced data-driven insights. The ongoing advancements in machine learning and AI are set to further push the boundaries of cancer treatment and surgical practices, aiming for more personalized and effective interventions. This research underlines the critical role of technological evolution in transforming cancer care, paving the way for future innovations in the Internet of Surgical Things.

## 6. Conclusions

In this article, we focused on identifying important biomarkers for each stage of breast cancer, with the goal of improving our understanding of the disease and potentially contributing to its early detection and targeted treatment. Our research involved the development of various methods to identify these biomarkers, using fold-change as a baseline for comparison. With fold-change, we identified up-regulated and down-regulated miRNAs for different stages of breast cancer. Additionally, we observed that the number of up-regulated genes decreases as the cancer stage progresses, while this trend does not hold for down-regulated genes.

Furthermore, we employed two popular feature extraction methods, NCA and MRMR, for identifying important biomarkers associated with different stages of breast cancers. Notably, the NCA algorithm was proven valuable in pinpointing stage-specific biomarkers for breast cancer, achieving high accuracy. However, the MRMR algorithm provided additional information about important biomarkers, allowing us to select common biomarkers for further investigation, ensuring their relevance and significance in the context of breast cancer staging.

The findings of this study have the potential to impact the field of oncology, offering insights into the disease’s progression and aiding in the development of personalized treatment strategies for patients at different stages. However, a limitation of the proposed approach is that blood samples cannot be included in the experimentation. This is because blood samples from patients exhibit the same miRNA composition for both normal tissues and cancer tissues, rendering them ineffective for pinpointing the most impactful miRNAs for breast cancer stage classification. Future tasks include identifying biomarkers for staging various cancers with limited data samples and clinically validating the significance of the biomarkers identified in this paper for cancer staging.

## Figures and Tables

**Figure 1 cancers-16-01864-f001:**
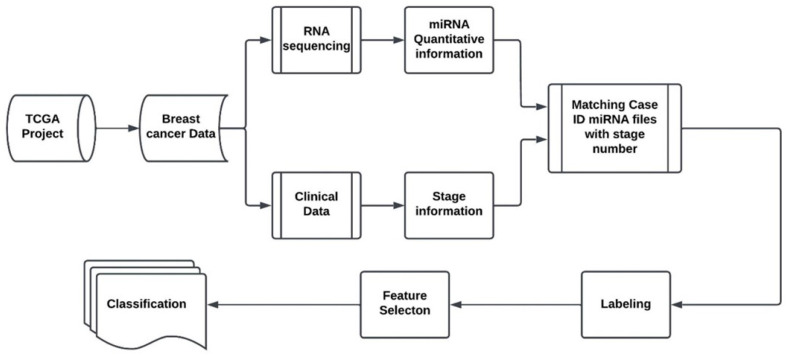
Proposed breast cancer stage classification system.

**Figure 2 cancers-16-01864-f002:**
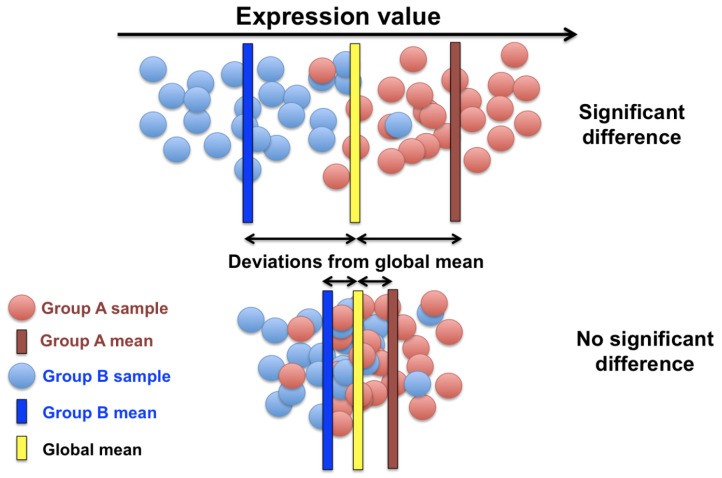
Significantly differentiated genes.

**Figure 3 cancers-16-01864-f003:**
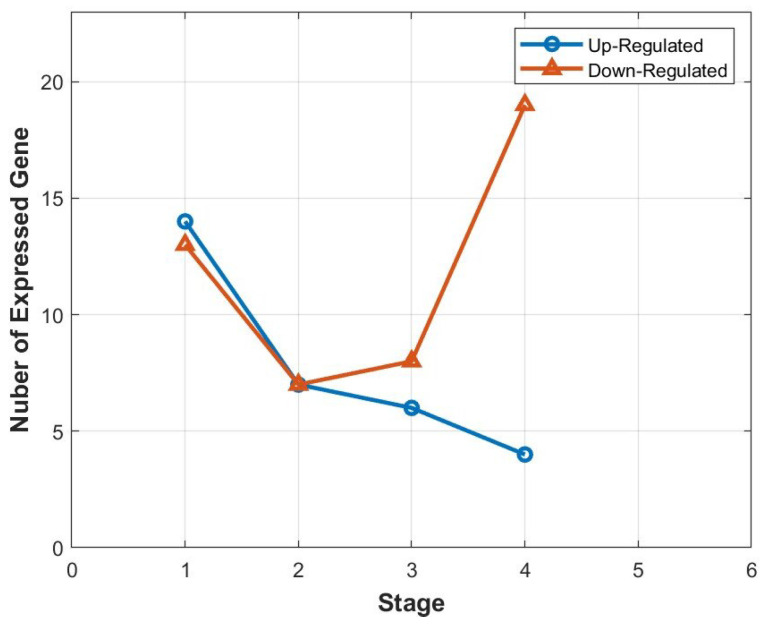
Correlation between the distinct stages and the number of expressed genes.

**Table 1 cancers-16-01864-t001:** Identification of up-regulated and down-regulated microRNAs in Stage 1.

MicroRNA	Mean Expression		Fold-Change	*p* Value	Adjusted p
	Tumor	Normal			
Up-regulated					
mir-122	3.7429	0.0459	81.6119	$4.1952\times 10^{-07}$	$1.8745\times 10^{-16}$
mir-4533	0.2623	0.0491	5.3343	$4.4909\times 10^{-04}$	$4.3840\times 10^{-05}$
mir-3156-2	2.3400	0.4547	5.1455	$9.2354\times 10^{-06}$	$8.2723\times 10^{-04}$
mir-490	0.2623	0.0491	5.3343	$4.4909\times 10^{-06}$	0.0272
mir-551a	1.1430	0.26161	4.3691	$1.9624\times 10^{-06}$	$1.9428\times 10^{-04}$
mir-3156-3	1.1186	0.2734	4.0911	$1.9151\times 10^{-05}$	0.0015
mir-383	5.0275	1.2439	4.0414	$1.1836\times 10^{-05}$	0.0010
mir-1295b	0.2954	0.0748	3.9449	0.00152451	0.0843
mir-323a	14.9393	3.8284	3.9022	$8.0076\times 10^{-05}$	0.0057
mir-137	1.5018	0.3899	3.8510	$1.6675\times 10^{-05}$	0.0013
Down-regulated					
mir-208b	0	0.5629	0	$5.4328\times 10^{-13}$	$7.8609\times 10^{-11}$
mir-206	3.9725	$2.3531\times 10^{02}$	0.0168	0	0
mir-133b	2.1395	$1.1519\times 10^{02}$	0.0185	$1.5579\times 10^{-91}$	$2.9305\times 10^{-89}$
mir-133a-2	5.5815	$2.5124\times 10^{02}$	0.0222	0	0
mir-133a-1	6.5369	$2.8983\times 10^{02}$	0.0225	0	0
mir-1-1	4.0193	$1.7249\times 10^{02}$	0.0233	$5.9569\times 10^{-116}$	$1.2449\times 10^{-113}$
mir-1-2	4.3460	$1.8343\times 10^{02}$	0.0236	$6.0448\times 10^{-118}$	$1.4212\times 10^{-115}$
mir-1269b	1.9498	26.7717	0.0728	$8.3273\times 10^{-24}$	$1.4239\times 10^{-21}$
mir-1911	0.1054	0.47800	0.2206	$5.6992\times 10^{-05}$	0.0042
mir-519a-1	1.7486	5.1012	0.3427	$1.4194\times 10^{-04}$	0.0098

**Table 2 cancers-16-01864-t002:** Identification of up-regulated and down-regulated microRNAs in Stages 2, 3, and 4.

Dysregulated miRNAs	Stage Two	Stage Three	Stage Four
Up-regulated	mir-1295b	mir-3156-3	mir-4533
	mir-122	mir-653	mir-551a
	mir-3156-3	mir-519a-1	mir-122
	mir-519a-1	mir-3156-2	mir-488
	mir-3156-2	mir-323a	-
	mir-3156-1	mir-3156-1	-
	mir-490	-	-
Down-regulated	mir-206	mir-208b	mir-133a-2
	mir-133b	mir-206	mir-1-1
	mir-133a-2	mir-133b	mir-133b
	mir-133a-1	mir-133a-2	mir-133a-1
	mir-1-2	mir-133a-1	mir-1-2
	mir-208b	mir-1-1	mir-1269b
	mir-1-1	mir-1-2	mir-206
	-	mir-1911	mir-208b
	-	-	mir-3156-2
	-	-	mir-3156-1

**Table 3 cancers-16-01864-t003:** Fold-change individual and combined accuracy.

Stage/Accuracy	Stage One	Stage Two	Stage Three	Stage Four
Up-regulated miRNAs	0.58	0.75	0.55	0.91
Down-regulated miRNAs	0.62	0.757	0.56	0.903
Combined accuracy	0.65	0.768	0.572	0.920

**Table 4 cancers-16-01864-t004:** Important genes for each of the four cancer stages identified with chi-square.

Stage One	Stage Two	Stage Three	Stage Four
mir-4659a	mir-7-2	mir-4524a	mir-4507
mir-124-3	mir-4712	mir-5193	mir-33a
mir-24-1	mir-3688-2	mir-3156-1	mir-548ay
mir-6761	mir-133a-2	mir-133a-2	let-7g
mir-208b	None	mir-1-1	mir-375

**Table 5 cancers-16-01864-t005:** Important genes for each of the four cancer stages identified with NCA.

Stage One	Stage Two	Stage Three	Stage Four
let-7a-2	let-7b	let-7a-1	let-7a-1
let-7b	let-7c	let-7a-2	let-7a-2
mir-10a	mir-10a	let-7a-3	let-7a-3
mir-10b	mir-10b	let-7b	let-7b
mir-143	mir-143	mir-101-1	mir-101-1
mir-148a	mir-148a	mir-101-2	mir-101-2
mir-182	mir-182	mir-103a-1	mir-103a-1
mir-21	mir-21	mir-103a-2	mir-103a-2
mir-22	mir-22	mir-10a	mir-10a
mir-30a	mir-30a	mir-10b	mir-10b
mir-375	None	mir-126	mir-126
None	None	mir-142	mir-142
None	None	mir-143	mir-143
None	None	mir-148a	mir-148a
None	None	mir-182	mir-182
None	None	mir-183	mir-183

**Table 6 cancers-16-01864-t006:** Important genes for each of the four cancer stages identified with MRMR.

Stage One	Stage Two	Stage Three	Stage Four
mir-3155a	mir-4676	mir-3155a	mir-6761
mir-1184-2	mir-8071-1	mir-4322	mir-345
mir-1290	mir-378h	mir-4417	mir-412
mir-1972-2	mir-4681	mir-4436a	mir-891a
mir-3119-2	mir-1253	mir-4502	mir-320c-1
mir-1184-1	mir-8079	mir-451b	mir-936
mir-378e	mir-8073	mir-5186	mir-378h
mir-4263	mir-5186	mir-526a-2	mir-4430
mir-4436b-2	mir-6089-1	mir-548ak	mir-7843
mir-4439	mir-3119-1	mir-548as	mir-5688

**Table 7 cancers-16-01864-t007:** Accuracy comparison of different models.

Stages/Algorithm	Stage One	Stage Two	Stage Three	Stage Four
NCA	0.94	0.947	0.953	0.983
FC	0.65	0.768	0.572	0.920
MRMR	0.87	0.881	0.916	0.931
Chi	0.76	0.83	0.878	0.861

**Table 8 cancers-16-01864-t008:** Common biomarkers identified by both NCA and MRMR.

Stage One	Stage Two	Stage Three	Stage Four
mir-10a	mir-133a-2	mir-1-1	mir-375
mir-208b	mir-148a	let-7b	let-7a-a
mir-24-1	mir-3688-2	mir-3156-1	mir-548ay
mir-6761	mir-133a-2	mir-133a-2	let-7g
mir-208b	None	mir-1-1	mir-375

**Table 9 cancers-16-01864-t009:** Algorithms’ accuracy using common features.

Stages/Algorithm	Stage One	Stage Two	Stage Three	Stage Four
Common Features	0.72	0.781	0.64	0.885

## Data Availability

Research data are publically available and can be accessed from the GDC data portal: https://portal.gdc.cancer.gov (accessed on 19 March 2024).

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
