# Peer review of "Identification of Gene Expression in Different Stages of Breast Cancer with Machine Learning"

_cancers, 2024, doi:10.3390/cancers16101864_

Round 1

Reviewer 1 Report (New Reviewer)

Comments and Suggestions for Authors

Though the work sounds interesting, few details need to be included to improve the conceptual clarity on the machine learning side. 

1. Enough details on the features employed and it basis behind selection is missing. 

2. Why SVM model is used for classification ? why not other models ? Also, what type of kernel functions were analysed as part of SVM. 

3. Comparision with SOTA Methods need to be provided. 

4. Research gaps and contributions need to be consolidated. 

5. Discussion section is very abstract. Kindly provide more inference on the features extracted, feature selection, and ML algorithms. 

Comments on the Quality of English Language

Moderate changes requried

Author Response

Thank you very much for providing us with valuable suggestions.

Though the work sounds interesting, more details need to be included to improve the conceptual clarity on the machine learning side. 

  1. Enough details on the features employed and its basis behind selection is missing. 

Reply: We added more details about the features selection methods employed in the revised paper.

  1. Why SVM model is used for classification? Why not other models? Also, what type of kernel functions were analyzed as part of SVM. 

Reply: The focus of the paper is not on identifying a better classifier, but on feature selection, for breast cancer stage classification. The Support Vector Machines (SVM) classifier was adopted in this study due to its suitability for binary classification. We compared SVM’s performance with those of KNN and Random Forest classifiers and tried different types of kernel functions for SVM such as Linear, Polynomial, and Radial Basis Function (RBF). We found that SVM with the Radial Basis Function (RBF) obtained the best results for the research task at hand. We added this to the revised paper.

  1. Comparison with SOTA Methods needs to be provided. 

Reply: In the revised paper, we include available SOTA articles in the revised article to include works on the stage cancer identification using microRNAs as biomarkers. However, we have not found much information about microRNA-based breast cancer stage classification using machine learning.

  1. Research gaps and contributions need to be consolidated.

Reply: We have addressed the research gaps and contributions in the introduction and literature review sections of the revised paper.

  1. The Discussion section is very abstract. Kindly provide more inference on the features extracted, feature selection, and ML algorithms. 

Reply: We have expanded this section to include a detailed analysis of the miRNA features extracted, the effectiveness of Neighborhood Component Analysis (NCA) and Minimum Redundancy Maximum Relevance (MRMR) algorithms for microRNA biomarker identification.  

Reviewer 2 Report (New Reviewer)

Comments and Suggestions for Authors

Major comments

1. You mentioned that you developed a model but you used SVM and other models that you didn't develop but were known. Correct this statement throughout the manuscript.

2. The introduction is too comprehensive. Compile the Related studies/Literature review as one section (From: In 2002, the first case of aberrant miRNA was identified .... to Yaqoob's group conducted a systematic review of the most nature-inspired algoritms, such as crows search....).

3. Clarify the number of data used and the disparity in the number of data obtained and the number of data used. For example, 716 samples were used out of 1207 samples. What happend to the rest? Please provide adequate explanation on that.

4. Where is the result to support this statement "Multiple machine learning approaches were applied for this purpose, and finally the SVM algorithm was selected due to its simplicity and satisfactory performance."

5. Briefly discuss Benjamini-Hochberg method.

6. The discussion section is too short. Expand to discuss the result outcome and limitation.

7. There is a fundamental problem with referencing scientific articles published from over 37 years ago. Are there not recent references to support your claims. References 2, 4, 6, 7, 8, and 9 should be replaced with more recent one.

Minor comments

1. Include the result in the abstract.

2. Re-arrange the keyword in ascending order.

Author Response

Reviewer #2

Thank you very much for providing us with valuable suggestions.

Major comments

  1. You mentioned that you developed a model, but you used SVM and other models that you didn't develop but were known. Correct this statement throughout the manuscript.

Reply: We corrected the statement in the revised paper.

  1. The introduction is too comprehensive. Compile the Related studies/Literature review as one section (From: In 2002, the first case of aberrant miRNA was identified .... to Yaqoob's group conducted a systematic review of the most nature-inspired algorithms, such as crows search....).

Reply: we divided the introduction into two sections, Introduction and Literature Review, following the suggestion.   

  1. Clarify the number of data used and the disparity in the number of data obtained and the number of data used. For example, 716 samples were used out of 1207 samples. What happened to the rest? Please provide an adequate explanation for that.

Reply: 1207 samples were extracted from the 1097 breast cancer patients, 113 of which are considered normal tissues samples taken from unaffected areas of the patient body. 111 diseased-tissue samples were obtained from patients who have stage-one breast cancer, 350 samples are associated with stage-two cases. 131 samples are correlated with stage-3 cancer, and 11 samples belong to stage 4. Samples that were labelled as Stage X were not included (which accounts for only five samples) in the original dataset.

Blood-specific samples were not used in the experimental study because blood samples from patients exhibit the same miRNA composition for both normal tissues and cancer tissues, rendering them ineffective for breast cancer stage classification. As a result, we used 716 tissue samples in our experiments.

  1. Where is the result to support this statement "Multiple machine learning approaches were applied for this purpose, and finally the SVM algorithm was selected due to its simplicity and satisfactory performance."

Reply: The Support Vector Machines (SVM) classifier was adopted in this study due to its suitability for binary classification. We compared SVM’s performance with those of KNN and Random Forest classifiers and tried different types of kernel functions for SVM such as Linear, Polynomial, and Radial Basis Function (RBF). We found that SVM with the Radial Basis Function (RBF) obtained the best results for the research task at hand. We added this to the revised paper.

  1. Briefly discuss Benjamini-Hochberg method.

Reply: The Benjamini-Hochberg method is a statistical approach designed to control the FDR in multiple hypothesis testing scenarios. This method involves ranking the p-values from multiple tests in ascending order, setting individual thresholds for each p-value based on its rank and the desired FDR level, and identifying the largest p-value that falls below its corresponding threshold as significant. We include discussion about Benjamini-Hochberg method in the revised paper.

  1. The discussion section is too short. Expand to discuss the result outcome and limitation.

Reply: We extended the discussion part of the article to include the outcomes and limitations of the proposed approach.

  1. There is a fundamental problem with referencing scientific articles published from over 37 years ago. Are there not recent references to support your claims. References 2, 4, 6, 7, 8, and 9 should be replaced with more recent ones.

Reply: We replaced old references with more recent ones.

Minor comments

  1. Include the result in the abstract.

Reply: We included results in the revised Abstract.

  1. Re-arrange the keyword in ascending order.

Reply: We make the correction.

Reviewer 3 Report (New Reviewer)

Comments and Suggestions for Authors

The manuscript presents an innovative study using machine learning to identify miRNA biomarkers in breast cancer. However, the manuscript could be enhanced by addressing certain areas, including methodological clarity, validation of findings, and broader comparison with existing research. Addressing these points will strengthen the manuscript's contribution to oncology and personalized medicine. Here are specific comments:

1. Clarity and Structure: The paper provides a comprehensive overview of the use of machine learning in identifying miRNA biomarkers for breast cancer stages. However, the clarity of certain sections could be improved for better readability. Specifically, the methodology section could benefit from a more detailed explanation of the algorithms used, including NCA and MRMR, and how they differ or complement each other in feature selection. Providing simplified examples or visual aids could enhance understanding.

2. Comparison with Existing Methods: The paper mentions the superiority of the proposed machine learning models over traditional Fold-Change methods. However, a more comprehensive comparison with other existing machine learning or bioinformatics approaches used in similar studies could provide a clearer picture of the proposed methods' advantages and limitations. For instance, compare with other machine learning models such as Random Forest, Gradient Boosting Machines, or other feature selection methods like Principal Component Analysis (PCA), Genetic Algorithms, or Recursive Feature Elimination (RFE), etc.

3. Statistical Analysis: The paper would benefit from a more detailed statistical analysis section. This should include a discussion on the handling of potential biases, the statistical significance of the findings, and how the models account for the high dimensionality and potential collinearity of miRNA expression data.

4. Limitations: The discussion on limitations is somewhat brief. A more thorough exploration of the study's limitations, including the potential for overfitting in machine learning models, the challenges of interpreting high-dimensional data, and the need for external validation of the findings, would provide a more balanced view.

Author Response

Reviewer #3

Thank you very much for providing us with valuable suggestions.

Comments and Suggestions for Authors

The manuscript presents an innovative study using machine learning to identify miRNA biomarkers in breast cancer. However, the manuscript could be enhanced by addressing certain areas, including methodological clarity, validation of findings, and broader comparison with existing research. Addressing these points will strengthen the manuscript's contribution to oncology and personalized medicine. Here are specific comments:

  1. Clarity and Structure: The paper provides a comprehensive overview of the use of machine learning in identifying miRNA biomarkers for breast cancer stages. However, the clarity of certain sections could be improved for better readability. Specifically, the methodology section could benefit from a more detailed explanation of the algorithms used, including NCA and MRMR, and how they differ or complement each other in feature selection. Providing simplified examples or visual aids could enhance understanding.

Reply: We added more details about the NCA and MRMR with mathematics description in the revised article.

2.Comparison with Existing Methods: The paper mentions the superiority of the proposed machine learning models over traditional Fold-Change methods. However, a more comprehensive comparison with other existing machine learning or bioinformatics approaches used in similar studies could provide a clearer picture of the proposed methods' advantages and limitations. For instance, compare with other machine learning models such as Random Forest, Gradient Boosting Machines, or other feature selection methods like Principal Component Analysis (PCA), Genetic Algorithms, or Recursive Feature Elimination (RFE), etc.

Reply: We made the correction in the revised article ‘s on your suggestions. Because PCA does not tell which miRNA contributes for cancer stage classification, it is not a preferred method for features selections.

We compared Random Forest and Chi-Square methods with the FC method (the baseline),and found out that they performed worse than the baseline method, therefore we did not report these results in the paper. We included this discussion in the revised article.

3.Statistical Analysis: The paper would benefit from a more detailed statistical analysis section. This should include a discussion on the handling of potential biases, the statistical significance of the findings, and how the models account for the high dimensionality and potential collinearity of miRNA expression data.

Reply: We include statistical results in the results and discussion.

  1. Limitations: The discussion on limitations is somewhat brief. A more thorough exploration of the study's limitations, including the potential for overfitting in machine learning models, the challenges of interpreting high-dimensional data, and the need for external validation of the findings, would provide a more balanced view.

Reply: We include the limitations and challenges in the discussion part.

Reviewer 4 Report (New Reviewer)

Comments and Suggestions for Authors

This paper studies Identification of Gene Expression in Different Stages of Breast Cancer with Machine Learning.  Two main feature-selection techniques have been used, mainly Neighborhood Component Analysis and Minimum Redundancy Maximum Relevance, to identify the most discriminant and relevant miRNAs for their up-regulated and down-regulated states. These miRNAs are then validated as biological identifiers for each of the four cancer stages in breast tumors. Both machine learning algorithms yield performance scores that are significantly higher than the traditional Fold-Change approach. Some weaknesses should be addressed in this paper. Therefore, I suggest the authors resubmit it after a major revision. My suggestions are as follows:

1. The introduction would benefit from further elaboration on the investigated problem. It is recommended that the authors incorporate additional details, particularly relevant statistics, to bolster the assertion of the problem. Including global statistics would strengthen the scientific evidence supporting their statements, thereby enriching the introduction.

2. At the end of the introduction, it is essential to provide a clear outline of the paper's structure to guide the reader through the upcoming sections.

3. Please increase the quality of the figure1.

4. When referring to a flowchart in Figure 1, you'll want to provide a clear and concise explanation of what the flowchart represents and how it contributes to understanding the topic or process discussed in the paper.

5. Divide introduction into two parts of introduction and literature review.

6. Please include the following relevant MDPI recent Machine learning healthcare papers as well:

- Identification of Breast Cancer Metastasis Markers from Gene Expression Profiles Using Machine Learning Approaches. Genes. 2023 Sep 20;14(9):1820.

- Machine learning methods for cancer classification using gene expression data: a review. Bioengineering. 2023 Jan 28;10(2):173.

A novel machine learning approach combined with optimization models for eco-efficiency evaluation. Applied Sciences. 2020 Jul 28;10(15):5210.

-  Analyzing RNA-seq gene expression data using deep learning approaches for cancer classification. Applied Sciences. 2022 Feb 11;12(4):1850.

-  Developing a novel integrated generalised data envelopment analysis (DEA) to evaluate hospitals providing stroke care services. Bioengineering. 2021 Dec 10;8(12):207.

-  Molecular classification models for triple negative breast cancer subtype using machine learning. Journal of Personalized Medicine. 2021 Sep 1;11(9):881.

- A novel hybrid parametric and non-parametric optimisation model for average technical efficiency assessment in public hospitals during and post-COVID-19 pandemic. Bioengineering. 2021 Dec 27;9(1):7.

7. The research issue must be articulated in a clear and concise manner to enhance comprehensibility.

8. The authors ought to offer deeper managerial insights derived from the research findings and effectively communicate the practical implications for managers and organizations. This ensures actionable guidance based on the study's outcomes.

9. Figure 2 (Significantly differentiated genes.) needs more clarification. In particular, lines 203-204: "The P-Value is also used as a statistical indicator for the likelihood of a gene to be deferentially expressed in this method" Why? Please explain more.

10. You need to add more content in the discussion parts as well. Research gap, future studies and the comparison of other ML algorithms are my suggestion to add in this part.

Comments on the Quality of English Language

Extensive editing of English language required

Author Response

Reviewer #4

Thank you very much for providing us with valuable suggestions.

Comments and Suggestions for Authors

This paper studies Identification of Gene Expression in Different Stages of Breast Cancer with Machine Learning.  Two main feature-selection techniques have been used, mainly Neighborhood Component Analysis and Minimum Redundancy Maximum Relevance, to identify the most discriminant and relevant miRNAs for their up-regulated and down-regulated states. These miRNAs are then validated as biological identifiers for each of the four cancer stages in breast tumors. Both machine learning algorithms yield performance scores that are significantly higher than the traditional Fold-Change approach. Some weaknesses should be addressed in this paper. Therefore, I suggest the authors resubmit it after a major revision. My suggestions are as follows:

  1. The introduction would benefit from further elaboration on the investigated problem. It is recommended that the authors incorporate additional details, particularly relevant statistics, to bolster the assertion of the problem. Including global statistics would strengthen the scientific evidence supporting their statements, thereby enriching the introduction.

Reply: we divided the introduction into two sections, Introduction and Literature Review in the revised paper. In addition, we added more details on the scientific evidence of microRNAs as biomarkers for breast cancer stage identification.

  1. At the end of the introduction, it is essential to provide a clear outline of the paper's structure to guide the reader through the upcoming sections.

      Reply: We included the outline of the remaining sections in the revised paper.

  1. Please increase the quality of Figure 1.

      Reply: We fixed Figure 1.

  1. When referring to a flowchart in Figure 1, you'll want to provide a clear and concise explanation of what the flowchart represents and how it contributes to understanding the topic or process discussed in the paper.

      Reply: We expanded the description related to Figure 1 in the revised paper.

  1. Divide introduction into two parts of introduction and literature review.

      Reply: As has been mentioned, we divided the introduction into introduction and literature review following the suggestion.

  1. Please include the following relevant MDPI recent Machine learning healthcare papers as well:

-     Identification of Breast Cancer Metastasis Markers from Gene Expression Profiles Using Machine Learning Approaches. Genes. 2023 Sep 20;14(9):1820.

-     Machine learning methods for cancer classification using gene expression data: a review. Bioengineering. 2023 Jan 28;10(2):173.

-     A novel machine learning approach combined with optimization models for eco-efficiency evaluation. Applied Sciences. 2020 Jul 28;10(15):5210.

-     Analyzing RNA-seq gene expression data using deep learning approaches for cancer classification. Applied Sciences. 2022 Feb 11;12(4):1850.

-     Developing a novel integrated generalized data envelopment analysis (DEA) to evaluate hospitals providing stroke care services. Bioengineering. 2021 Dec 10;8(12):207.

-     Molecular classification models for triple negative breast cancer subtype using machine learning. Journal of Personalized Medicine. 2021 Sep 1;11(9):881.

- A novel hybrid parametric and non-parametric optimization model for average technical efficiency assessment in public hospitals during and post-COVID-19 pandemic. Bioengineering. 2021 Dec 27;9(1):7.

      Reply: We include these articles in the revised manuscript.

7.The research issue must be articulated in a clear and concise manner to enhance comprehensibility.

      Reply: We revised the article based on this comment.  

8.The authors ought to offer deeper managerial insights derived from the research findings and effectively communicate the practical implications for managers and organizations. This ensures actionable guidance based on the study's outcomes.

      Reply: We have revised the article to include deeper managerial insights derived from our research findings. We have also elaborated on the practical implications for managers and organizations to ensure the guidance we provide is actionable and directly applicable. These enhancements aim to better bridge the gap between theory and practical application, offering clear benefits to practitioners in the field.

9.Figure 2 (Significantly differentiated genes.) needs more clarification. In particular, lines 203-204: "The P-Value is also used as a statistical indicator for the likelihood of a gene to be deferentially expressed in this method" Why? Please explain more.

      Reply: In our analysis, we employ the P-value alongside Fold-Change to assess the statistical significance of differences in gene expression between two conditions. The Fold-Change itself provides a measure of the magnitude of expression difference, but it does not indicate the statistical reliability of this difference. By incorporating the P-value, we aim to determine whether the observed changes in gene expression are likely to be due to random variation or are statistically significant. Specifically, the P-value helps in testing the null hypothesis that there is no difference in gene expression between the treatment and control groups. A low P-value (typically less than 0.05) indicates that the observed Fold-Change is unlikely to have occurred by chance, thereby supporting the alternative hypothesis that there is a significant difference in expression. This statistical approach ensures that the identified differentially expressed genes are not only different in terms of Fold-Change but also significant statistically, minimizing the risk of false positives. For instance, in Figure 2 where significantly differentiated genes are plotted, each gene's Fold-Change is calculated as mentioned. The corresponding P-value for each gene's expression difference is then computed to assess the likelihood that such a difference arose by chance. This method allows us to confidently identify genes that are differentially expressed under the experimental conditions studied, providing robust support for further biological interpretation and validation.

  1. You need to add more content to the discussion parts as well. Research gap, future studies and the comparison of other ML algorithms are my suggestion to add in this part.

      Reply: We revised the article based on this comment.  

Round 2

Reviewer 1 Report (New Reviewer)

Comments and Suggestions for Authors

Authors have addressed my suggestions. 

Comments on the Quality of English Language

Minor changes

Reviewer 4 Report (New Reviewer)

Comments and Suggestions for Authors

Authors answers all of my comments and this version is available for the publication.

Comments on the Quality of English Language

Minor English Check is required.

This manuscript is a resubmission of an earlier submission. The following is a list of the peer review reports and author responses from that submission.

Round 1

Reviewer 1 Report

Comments and Suggestions for Authors

This research aims to use machine learning to find effective miRNAs for classification of breast cancer stages. The authors focused on the identification of biomarkers in each stage of Breast Cancer, aiming to improve our understanding of the disease and potentially contribute to its early detection and targeted treatment. Using NCA and MRMR they identified deregulated miRNAs for different stages of breast cancer, and stated their future tasks is to identify biomarkers for staging different cancers with few data samples.

The authors attempt to build a tissue agnostic, miRNA based classifier ("We did not separate samples that were extracted from blood or tissue"). However, miRNA expression is tissue specific, and comparing blood expression in stage one to tissue expression in another stage has no clinical significance. Can the authors elaborate how they plan to compare tissue expression vs blood expression without having the basal expression of that miRNA in a specific tissue? A specific miRNA expression can be 100X in the breast compared to the blood. In addition, please provide a supplementary table of the location of each of your samples- blood or tissue.

Cancer staging goes from 1 (small, localized tumor) to 4 (metastatic disease). Some classification systems include 0 (in-situ lesion, premalignant). The authors mentioned that they included 5 stages- what did they mean by that? There is no stage 5 in any cancer.

“1207 samples were extracted from the 1097 breast cancer patients, 113 of which are considered normal tissues samples taken from unaffected areas of the patient body. 111 diseased-tissue samples were obtained from patients who have stage-one breast cancer, 350 samples are associated with stage-two cases. 131 samples are correlated with stage-3 cancer, and 11 samples belong to stage 4. Finally, only 5 samples are categorized as stage 5”- that sums up to 721 samples- which group did the other samples originate from?

In the text the authors mention- “Table 1 shows the five different stages of the breast cancer miRNA expressions that have been revealed to be the most influential biomarkers in detecting each tumor stage”. The header of table 1 is “Identification of Upregulated and Downregulated MicroRNAs in Stage 1”- Is table 1 specific to stage 1 or it contains additional cancer stages?

Table 1 includes "Adjusted p"- how was that calculated? please explain how in table 1 some of the cases the adjusted p in smaller then the p value (miR-122)

What did the authors aim to convey in table 2? The manuscript doesn’t contain any reference for table 2, and the table itself is very unexplanatory.

Table 3- “The accuracy of the up-regulated and down-regulated genes in identifying their respective corresponding stages are shown in Table 3”- provide an explanation on how the accuracy was calculated.

Do you compare each cancer stage to the normal breast tissue? Else? Please elaborate.

The results of the FC comparison and the NCA/MRMR should be presented together so that the authors will have better way to compare the methods.

Comments on the Quality of English Language

“rental cells carcinoma”

"influnential biomarkers"

There are several other examples

Reviewer 2 Report

Comments and Suggestions for Authors

The primary objective of the research paper is to develop a machine learning model that leverages an array of miRNAs in metastatic tissue samples from 1097 patients with varying stages of breast cancer. The authors employ two main feature-selection techniques, namely Neighborhood Component Analysis and Minimum Redundancy Maximum Relevance, to identify the most pertinent and informative miRNAs in their up-regulated and down-regulated states. The results of the study indicate that the proposed machine learning algorithms outperform the traditional Fold-Change approach in terms of performance scores. Below are some my concern:

1.      Generally, I recommend revising the English writing to correct some grammar errors and typos in the document.

2.      In the introduction, the need to justify why the researchers in the literature used  feature selection techniques to solve the gene selection problem.

3.      The authors need to justify why they use Neighborhood Component Analysis and Minimum Redundancy Maximum Relevance, rather than other feature selection techniques to solve the gene selection problem.

4.      The recent relevant research papers should be considered in the literature section and can use in the comparisons like: An enhanced binary Rat Swarm Optimizer based on local-best concepts of PSO and collaborative crossover operators for feature selection, A review on nature-inspired algorithms for cancer disease prediction and classification, Hybrid Feature Selection Techniques Utilizing Soft Computing Methods for Cancer Data, Novel machine learning approach for classification of high-dimensional microarray data, Binary Horse herd optimization algorithm with crossover operators for feature selection.

5.      The authors need to highlight the research gap at the end of the literature review section.

6.      The authors need to add more analysis for the results obtained by their method in the comparison section.

7.      What is the percentage of training and testing sets?

8.      Conclusion is not at all satisfactory. It should be concise, and your contribution and novelty should be claimed.

9.      Lastly, the paper does not address potential limitations or challenges associated with the proposed algorithm.

Comments on the Quality of English Language

Minor editing of English language required

Reviewer 3 Report

Comments and Suggestions for Authors   GENERAL COMMENTS I was glad to review the article entitled "Identification of Gene Expression in Different Stages of Breast Cancer with Machine Learning". This manuscript aims to use machine learning to find effective miRNAs for the classification of breast cancer stages. Neighborhood Component Analysis (NCA) and Minimum Redundancy Maximum Relevance (MRMR) algorithms were used to identify the most relevant miRNA for each stage. The topic is original and relevant to the field. There is limited information on this topic in the literature. This article is well written and important as the findings of this study have the potential to significantly impact the field of oncology, providing crucial insights into the disease’s progression and aiding in the development of personalized treatment strategies for patients at different stages. There are no further improvements regarding the methodology.
The conclusions are consistent with the evidence and arguments presented as well as summarize the main point of this article. 
References are up-to-date and appropriate
Tables and figures are well formatted and make the study easy to follow   MINOR REVISION   1) "HER2 is an established prognostic and predictive marker for patients with invasive breast cancer. The clinical and biological significance of HER2 overexpression in patients with ductal carcinoma in situ (DCIS) remains poorly defined."   Add this important information and make a brief discussion on the clinical significance of HER2 expression in DCIS Consider citing: https://pubmed.ncbi.nlm.nih.gov/36352293/   2)“In the last few years, technological developments in the medical/surgical field have been rapid and are continuously evolving. One of the most revolutionizing breakthroughs was the introduction of the IoT concept within the medical and surgical practice.”

Add this information in the discussion section and explain the role of IoT in Machine Learning related to breast cancer

Consider citing the article on the Internet of surgical things

https://pubmed.ncbi.nlm.nih.gov/35746359/